# Gene Expression and Interaction Analysis of *FsWRKY4* and *FsMAPK3* in *Forsythia suspensa*

**DOI:** 10.3390/plants12193415

**Published:** 2023-09-28

**Authors:** Xinjie Tan, Jiaxi Chen, Jiaqi Zhang, Guangyang Guo, Hongxiao Zhang, Xingli Zhao, Shufang Lv, Huawei Xu, Dianyun Hou

**Affiliations:** College of Agriculture, Henan University of Science and Technology, Luoyang 471032, China

**Keywords:** *Forsythia suspensa*, *FsWRKY4*, *FsMAPK3*, gene expression, interaction

## Abstract

*Forsythia suspensa* is a deciduous shrub that belongs to the family Myrtaceae, and its dried fruits are used as medicine. *F. suspensa* contains several secondary metabolites, which exert pharmacological effects. One of the main active components is forsythin, which exhibits free radical scavenging, antioxidant, anti-inflammatory, and anti-cancer effects. Mitogen-activated protein kinase (MAPKs) can increase the activity of WRKY family transcription factors in a phosphorylated manner, thereby increasing the content of secondary metabolites. However, the mechanism of interaction between *MAPKs* and *WRKYs* in *F. suspensa* remains unclear. In this study, we cloned the genes of *FsWRKY4* and *FsMAPK3*, and performed a bioinformatics analysis. The expression patterns of FsWRKY4 and FsMAPK3 were analyzed in the different developmental stages of leaf and fruit from *F. suspensa* using real-time fluorescence quantitative PCR (qRT-PCR). Subcellular localization analysis of FsWRKY4 and FsMAPK3 proteins was performed using a laser scanning confocal microscope. The existence of interactions between FsWRKY4 and FsMPAK3 in vitro was verified by yeast two-hybridization. Results showed that the cDNA of *FsWRKY4* (GenBank number: OR566682) and *FsMAPK3* (GenBank number: OR566683) were 1587 and 522 bp, respectively. The expression of *FsWRKY4* was higher in the leaves than in fruits, and the expression of *FsMAPK3* was higher in fruits but lower in leaves. The subcellular localization results indicated that FsWRKY4 was localized in the nucleus and FsMAPK3 in the cytoplasm and nucleus. The prey vector pGADT7-FsWRKY4 and bait vector pGBKT7-FsMAPK3 were constructed and co-transferred into Y2H Glod yeast receptor cells. The results indicated that FsWRKY4 and FsMAPK3 proteins interact with each other in vitro. The preliminary study may provide a basis for more precise elucidation of the synthesis of secondary metabolites in *F. suspensa*.

## 1. Introduction

*Forsythia suspensa* is a deciduous shrub that belongs to the family Myrtaceae [1]. The dried fruits are used as medicine to clear heat, detoxify the body, reduce swelling, disperse knots, and reduce wind-heat [2]. It is mostly found in the Henan, Hebei, Shandong, Shaanxi, and Shanxi provinces [1]. *F. suspensa* is a traditional Chinese medicine that has high efficacy in the prevention and treatment of novel coronavirus pneumonia [3].

The development of forsythia mainly focuses on medicinal, edible, and cosmetic aspects. In terms of medicine, *F. suspensa* has the functions of liver protection and diuretic catheterization, and anti-inflammatory and analgesic properties [4]. In terms of consumption, *F. suspensa* can be processed into tea and has health benefits, and *F. suspensa* seeds can also be extracted into oil and have high nutritional value [4].

The phenylpropane pathway is rich in phenylpropanoids and natural compounds [5]. The phenylpropane metabolic pathway is one of the most important metabolic pathways in plants and can produce more than 8000 metabolites [6]. This situation is particularly evident in medicinal plants [7]. More than 70% of pharmaceutical products are based on plant metabolites, including lignans, flavonoids, coumarins, quinones, and lignans, which are derived from the phenylpropane pathway [8]. Phenylpropanoids are antioxidants, free radical scavengers, and anti-inflammatory and anti-cancer compounds [9]. The chemical compositions of forsythia are complex [10]. In the 2020 edition of the Chinese Pharmacopoeia, lignans, phenyl ethanol glycosides, forsythin, and forsythia glycoside A, which are produced through the phenylpropane pathway, are listed as the main active constituents of coniferin [11,12]. It includes *F. suspensa* ester glucoside and forsythia glucoside with good anti-influenza virus effect [13].

In most plants, the phenylpropane metabolic pathway starts with the production of phenylalanine by the manganate pathway and the glycolytic pathway [14]. The synthesis pathway of forsythia glycosides is regulated by the phenylpropane synthesis pathway [15]. Therefore, studying the phenylpropane pathway of forsythin has far-reaching implications in improving the utilization of *F. suspensa*.

The WRKY transcription factor was first reported in sweet potato in 1994 [16]; since, an increasing number of WRKY transcription factors have been identified in a variety of medicinal plants with the function of enhancing the content of secondary metabolites [17,18,19,20]. Current studies on the regulation of secondary metabolite production by WRKY transcription factors mainly focus on phenolics, alkaloids, and terpenoids [21]. For example, *ZoWRKY1* in ginger can regulate the production of 6-gingerol (phenylpropanoids) by up-regulating the *ZoPAL*, *ZoC4H*, and *Zo4CL* genes [22]. The overexpression of *VvWRKY2* in tobacco resulted in delayed xylem formation in tobacco stems and petioles and varying degrees of reduction in lignin content [23]. However, no relevant reports have been found regarding *F. suspensa*.

The MAPK cascade reaction is involved in the synthesis of plant secondary metabolites through phosphorylation with transcription factors [24]. For example, in *Astragalus*, the induced activation of *AmMAPK3* and *AmMAPK6* regulated the expression of *AmPAL* and *AmI3′H*, leading to a change in the content of hypromellose and calycin in vivo [25]. The activation of *AtMAPK4* by light led to the phosphorylated modification of the *AtMYB75* transcription factor and promoted the accumulation of anthocyanins [26]. In *Salvia miltiorrhiza*, two cascades, namely, SmMAPK3 and SmMAPK1, were highlighted as potentially being involved in the synthesis of phenolic acids and tanshinones [27].

WRKY transcription factors are localized in the nucleus, and the MAPK cascade can transmit external signals into the nucleus. These factors interact with MAPK to regulate the expression of the corresponding enzyme genes in response to biotic and abiotic stresses [28],e.g., the GhMAP3K15–GhMKK4–GhMPK6–GhWRKY59–GhDREB2 cascade reaction in cotton’s regular drought response [29]. In tobacco [30], SIPK can directly phosphorylate *NtWRKY1* as a downstream of SIPK and regulate the expression of cell death-related genes.

The multiple sequence alignment, phylogenetic analysis, and conserved motif construction of WRKY have been reported in our previous research [31]. In this study, the transcriptome analysis of *F. suspensa* showed that *FsWRKY4* and *FsMAPK3* may interoperate and mediate the phenanthrene pathway. The relationship between FsWRKY4 and FsMAPK3 proteins were explored by cloning to obtain the WRKY transcription factor *FsWRKY4* and its reciprocal gene *FsMAPK3* from *F. suspensa*. The results provide suggestions to effectively improve the extraction rate of *F. suspensa* glycosides as an index of potency. This work provides useful reference for in-depth study of the molecular mechanism of WRKY transcription factors in regulating the secondary metabolite synthesis pathway.

## 2. Results

### 2.1. Cloning of FsWRKY4 and FsMAPK3

The target fragments *FsWRKY4* and *FsMAPK3* were amplified by PCR and examined by 1% agarose gel electrophoresis (Figure 1A,B). The positive bacterial solution was selected and sent to the company for sequencing. The results indicated that the cDNA sequences of FsWRKY4 and FsMAPK3 were 1587 and 522 bp, respectively. The *FsWRKY4* and *FsMAPK3* gene sequence have been submitted to GenBank. The GenBank numbers of *FsWRKY4* and *FsMAPK3* were OR566682 and OR566683, respectively.

### 2.2. Spatiotemporal Expression Patterns of FsWRKY4 and FsMAPK3

The relative expression levels of *FsWRKY4* and *FsMAPK3* genes in different tissues at different times were investigated by qRT-PCR in the leaves and fruits of medicinal plants in June, July, August, and September. The induction of the expression levels of the target genes at four different developmental periods was differential, with *FsWRKY4* in fruit in June and *FsMAPK3* in leaves in June as controls. The expression of both target genes in leaves reached the maximum in July and then gradually decreased. The expression of *FsWRKY4* and *FsMAPK3* was always higher in leaves than in fruits of *F. suspensa* (Figure 2A,B).

### 2.3. Subcellular Localization Analysis of FsWRKY4 and FsMAPK3 Proteins

The recombinant expression vectors pCAMBIA 1300-FsWRKY4-GFP and pCAMBIA 1300-FsMAPK3-GFP were constructed and transfected into an *Agrobacterium tumefaciens* GV3101 receptor state to infect the lower epidermis of tobacco leaves. Laser confocal microscopy showed the results in Figure 3. The pCAMBIA 1300-GFP vector was used as a control. pCAMBIA 1300-FsWRKY4-GFP was transiently expressed into tobacco leaves, and the fluorescent DNA stain DAPI was used to indicate the nuclear region. The colocalization of the GFP and DAPI signals indicated that FsWRKY4 was specifically localized within the nucleus, consistent with the prediction of software (Cell-Ploc 2.0) (Figure 3A). The luminescent location of the pCAMBIA 1300-FsMAPK3-GFP vector was observed to be located in the cytoplasm and nucleus of tobacco, with pCAMBIA 1300-UGPase-GFP as a control. The FsMAPK3 protein was expressed in the cytoplasm and nucleus (Figure 3B).

### 2.4. Yeast Toxicity Testing

During the yeast two-hybrid assay, the transferred fragments may become toxic to yeast when expressed in yeast, thereby affecting the assay. The recombinant decoy vector containing the upstream activation sequence needs to be tested for yeast toxicity prior to the two-hybrid assay. The yeast two-hybrid recombinant decoy vectors pGBKT7-FsMAPK3 and pGBKT7 were transferred into the yeast Y2H Gold receptor state. They were coated on a SD/-Trp medium and incubated at 29 °C in inverted darkness for 48–96 h to compare colony growth. The results are shown in Figure 4A,B. Colony density and size were not significantly different among the samples. The decoy protein was proved to be non-toxic and could be tested in the next step.

### 2.5. Yeast Two-Hybrid Recombinant Plasmid Self-Activation Assay

The yeast two-hybrid plasmids were analyzed according to pGADT7-T×pGBKT7-53 (positive control), pGADT7-T×pGBKT7-Lam (negative control), pGADT7-T×pGBKT7-Lam (negative control), pGADT7×pGBKT7 (empty plasmid control), pGADT7-FsWRKY4×pGBKT7 (recombinant prey vector test group), and pGADT7×pGBKT7-FsMAPK3 (recombinant decoy vector test group) combinations to co-transform the yeast Y2H Gold strain and verify whether the yeast two-hybrid recombinant plasmids have a self-activating ability. According to gradient dilution, the colonies were spot-seeded on corresponding media containing 130 μg/mL AbA.

The spot seeding is shown in Figure 4C. The positive control Y2H[pGADT7-T×pGBKT7-53] combination grew on DDO, DDO/X, QDO/X, and QDO/X/A media. Blue colonies were grown on a medium containing X-α-gal. The negative control Y2H[pGADT7-T×pGBKT7-Lam] only grew colonies on DDO and DDO/X media, but the colonies grown on DDO/X were not blue. The growth of the empty vector plasmid combination Y2H[pGADT7×pGBKT7], the recombinant prey vector test group Y2H[pGADT7-FsWRKY4×pGBKT7], and the recombinant decoy vector test group Y2H[pGADT7×pGBKT7-FsMAPK3] on the defective medium was the same as the negative control. The recombinant plasmids pGADT7- FsWRKY4 and pGBKT7-FsMAPK3 can be inhibited at 130 μg/mL AbA, as they have a heavy self-activation ability, and can be further used for yeast two-hybrid assay.

### 2.6. Analysis of FsWRKY4 and FsMAPK3 Protein Interactions

After the verification of protein toxicity and self-activation, excluding other factors, the protein interactions of FsWRKY4 and FsMAPK3 were analyzed. The recombinant plasmid combination pGADT7-FsWRKY4×pGBKT7-FsMAPK3 was co-transformed to the yeast Y2H Gold strain. After the completion of transformation, they were coated with defective media DDO and QDO and incubated at 29 °C in inverted darkness for 48–96 h. The bacterial broth was spot seeded on the DDO, DDO/X, QDO/X, and QDO/X/A media, according to gradient dilution. As shown in Figure 5, all colonies grew on DDO, DDO/X, QDO/X, and QDO/X/A media, and the colonies grown on DDO/X, QDO/X, and QDO/X/A media were blue. The DNA-specific binding domain (BD) and the transcriptional activation domain (AD) formed a complete transcriptional activator and activated four selection marker genes (AUR1-C (AbA gold tamarin resistance screen), MEL1 (X-α-gal blue-white spot screen), HIS3 (His histidine nutritional screen), and ADE2 (Ade adenine nutritional screen)) with two to be tested for protein binding. The findings indicated the presence of an interaction between the FsWRKY4 and FsMAPK3 proteins.

## 3. Discussion

China is rich in *F. suspensa*, which is a clinically used herbal medicine because of its abundant resources, complex composition, and diverse biological activities [32]. It can be combined with a variety of herbal medicines to produce a wide range of proprietary Chinese medicines with great prospects for development and utilization [33]. In addition to its medicinal value, it has economic, ecological, and ornamental values, as well as other applications [34].

At present, studies on *F. suspensa* have focused on chemical composition and extraction, as well as pharmacological activity; however, few works have reported on the metabolic pathways of the active ingredients of *F. suspensa* [35]. The metabolic pathway analysis of the active ingredients of *F. suspensa* can be used to investigate their mechanism of action and provide a basis for clinical application.

In recent years, the research of *WRKY* and *MAPK* genes has attracted much attention. Many studies have been conducted on the respective functions of *WRKY* and *MAPK* genes and their interactions [36,37,38,39]. All these studies provide references and ideas for our research.

The mitogen-activated protein kinase cascade reaction is a highly conserved signaling system, which is widely found in eukaryotic cells such as plants and animals [40]. In plant cells, MAPK, MAPK kinase, MAPKK kinase and MAPKKK kinase form the MAPK cascade, which plays an important role in plant physiological processes and the oxidative stress response [41] to biotic [42,43] and abiotic stresses [44,45]. The role of enzymatic and non-enzymatic antioxidants as integrating factors in multiple signaling cascades can be used to prevent oxidative damage under unfavorable abiotic stress conditions [46]. The phosphorylation and activation of MPAPKs are used to transform stimuli or regulate stress systems in response to stress and non-stress [47]. 

The MAPK signaling pathway has been reported to enhance secondary metabolite content by activating WRKY family transcription factors in a variety of plants. For example, in the cotton group IIc, WRKY transcription factors regulate flavonoid biosynthesis by activating GhMKK2-mediated pathways [48]. In *Arabidopsis*, an interplay exists between AtWRKY3 and AtMPK3/AtMPK6, and regulates the accumulation of phytoalexins biosynthesis [49], which provides a reference for us to study the involvement of *FsMAPK3* in the secondary metabolites of *F. suspensa*. The sustained activation of AtMAPK3/AtMAPK6 by AtWRKY affects the biosynthesis of myricetin and the accumulation of the antimicrobial compound pipecolic acid [50], and indicated that AtMAPK3/AtMAPK6 and AtWRKY were interacting proteins, which was consistent with the results of our yeast two-hybrid study of *F. suspensa*.

There have been many reports about the interaction of MAPK and WRKY. In *A. thaliana*, a subset of VQ-motif-containing proteins (VQPs) were phosphorylated using the MAPKs MPK3 and MPK6, which renamed MPK3/6-targeted VQPs (MVQs) [51]. Through the yeast two-hybrid technique, it is proved that MVQs and WRKY are interacting with each other [51]. In tobacco, through interaction with the MAPK cascade, WRKYs can regulate and modulate plant defense against whiteflies, which indicates that MAPK and WRKYs are interacting proteins [52]. The above studies are consistent with our findings, suggesting that MAPKs and WRKYs have an interaction. In the study, the expression of FsWRKY4 and FsMAPK3 were analyzed in the leaves and fruits of *F. suspensa* during the different developmental stages. Similar studies have been reported on *osmanthus fragrans* [53] and *salvia miltiorrhiza* [54].

In this study, we screened the transcriptomic data for WRKY transcription factors that may be involved in the phenol propane pathway in *F. suspensa*. We further investigated the interactions between FsWRKY4 and FsMAPK3 using qRT-PCR, subcellular localization, and yeast two-hybridization. The results provide a reference for the in-depth analysis of the molecular mechanism of WRKY transcription factors in regulating the biosynthesis pathway of secondary metabolites in *F. suspensa*.

## 4. Materials and Methods

### 4.1. Acquisition of Test Materials

*F. suspensa* plants grown in Yiyang County, Luoyang City, Henan Province, China were used as the study material. The growth of *F. suspensa* plants was performed under normal water and fertilizer conditions. Single *F. suspensa* plants were labeled, and the leaves and fruits were picked on the 10th of every month from June to September and were checked for pests and diseases, prior to their use as plant materials. The leaves and fruits were snap frozen in liquid nitrogen and labeled. 

The plant materials were stored in the Luoyang Engineering Research Center of Breeding and Utilization of Dao-di Herbs. The herbarium number for the *F. suspense* plant is LYLQ001.

### 4.2. RNA Isolation

Approximately 0.07 g of the fruit and leaves of *F. suspensa* were weighed for RNA extraction. The samples were thoroughly ground in liquid nitrogen, and the total RNA was obtained from plant leaves and fruits according to the steps of the RNE35 Broad Spectrum Plant RNA Extraction Kit (Beijing, China, Noble).

### 4.3. Cloning of FsWRKY4 and FsMAPK3 Genes in F. suspensa

The sequences of TRINITY_DN14046_c0_g1 and TRINITY_DN4223_c0_g1 genes in the transcriptome data of conidia were used as a reference. Based on the homology between the two sequences and the sequence in *A. thaliana*, TRINITY_DN14046_c0_g1 and TRINITY_DN4223_ c0_g1 were named as *FsWRKY4* and *FsMAPK3*, respectively. The primer sequences are shown in Table 1. Approximately 50 μL of the polymerase chain reaction system included 25 μL of 2×PrimerStar Max DNA Polymerase, 2 μL of 2.5 M forward primer, 2 μL of 2.5 M reverse primer, 2 μL of cDNA and 19 μL of sterile water. The amplification reaction conditions were as follows: pre-denaturation at 98 °C for 10 s, followed by denaturation at 94 °C for 15 s, annealing at 58 °C for 30 s, extension at 72 °C for 1 min, and final extension at 72 °C for 5 min. After amplification, the correct band length was judged by 1% agarose gel electrophoresis, followed by purification, recovery, and ligation to the pMD-18T vector. Finally, the sequence was transformed into E. *scherichia* coli receptor DH5α and the positive clones were sequenced. The correctly sequenced and extracted plasmids were used in the next step of the experiment.

### 4.4. Quantitative Real-Time Polymerase Chain Reaction Analysis

The cDNA was prepared according to the method provided by Shanghai Yi Sheng Company. Primers were designed to detect the expression levels of *FsWRKY4* and *FsMAPK3* with the reference gene UKN1 by using Primer Premier 5.0 software (Table 2).

The relative expression of target genes was detected by the SYBR qPCR Master Mix (Novozymes, Nanjing, China) and the Roche Lightcycler 96 fluorescent quantitative polymerase chain reaction instrument. Each reaction was repeated three times, and the 2-ΔΔCt method was used to calculate the expression of *FsWRKY4* and *FsMAPK3*, which were detected in the leaves and fruits of forsythia at different developmental stages.

### 4.5. Subcellular Localization of FsWRKY4 and FsMAPK3 

The subcellular localized recombinant vector pCAMBIA 1300-FsWRKY4/FsMAPK3-GFP was constructed by homologous recombination. Homologous primers were designed using CE Design software (V.1.04) in combination with the target gene open reading frame (with the stop codon removed). The primer sequences are shown in Table 3. 

The pCAMBIA 1300-GFP vector was cut using Kpn I and Xba I enzymes (Takara, Kusatsu, Japan) and ligated using homologous recombinase (Novozymes, Nanjing, China) to obtain pCAMBIA 1300-FsWRKY4/FsMAPK3-GFP. The GFP vector was ligated using homologous recombinase (Novozymes, Nanjing, China) to obtain pCAMBIA1300-FsWRKY4/FsMAPK3-GFP.

The recombinant plasmid was transformed into *A. tumefaciens* GV3101, allowed to infiltrate the lower epidermis of the tobacco, and incubated in the dark for 2 d. The subcellular localization of FsWRKY4 and FsMAPK3 proteins was observed by confocal laser microscopy with excitation light sources of 488 and 583 nm, respectively.

### 4.6. Yeast Two-Hybrid

The yeast two-hybrid recombinant prey vector pGADT7-FsMAPK3 and recombinant decoy vector pGBKT7-FsWRKY4 were constructed using the homologous recombination method. Homologous primers were designed using CEII Design software in combination with the target gene open reading frame (Table 4). 

The pGADT7 and pGBKT7 vectors were cut using NdeI and BamHI enzymes (Takara). The constructed recombinant plasmids were transferred into Y2H Gold yeast receptor cells. After correct sequencing, a yeast toxicity assay and self-activation assay were performed. Finally, the plasmids were transferred into Y2H Gold yeast receptor cells for the yeast two-hybrid assay to verify protein interactions.

### 4.7. Statistic Analysis

The expression changes and significance analysis of FsWRKY4 and FsMAPK3 in different developmental stages was performed by SPSS 5.0 software.

## 5. Conclusions

In this study, WRKY transcription factors that may be involved in the phenylpropane pathway of *F. suspensa* were screened from the transcriptomic data. The existence of interactions between FsWRKY4 and FsMAPK3 in *F. suspensa* were explored by a bioinformatics analysis, a qRT-PCR assay, the subcellular localization of the proteins, and the yeast two-hybrid technique. The results provide support for further research on the regulatory mechanism of FsWRKY4 and FsMAPK3 on the active ingredients.

## Figures and Tables

**Figure 1 plants-12-03415-f001:**
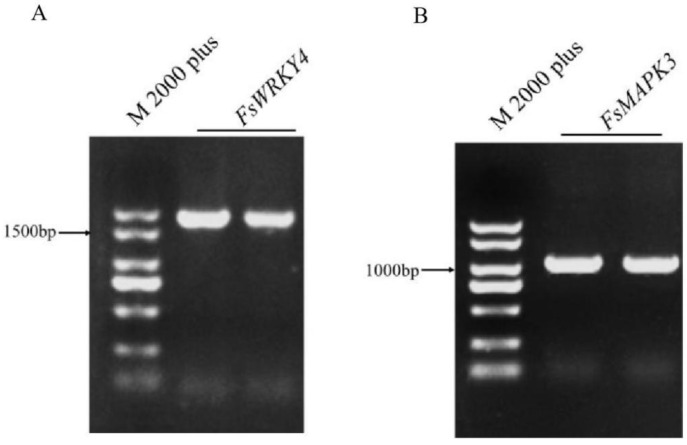
cDNA fragment of *FsWRKY4* and *FsMAPK3*. “M 2000 plus” denotes DNA marker, which is used to pinpoint DNA band sizes. (**A**) cDNA fragment of *FsWRKY4*; (**B**) cDNA fragment of *FsWRKY4*.

**Figure 2 plants-12-03415-f002:**
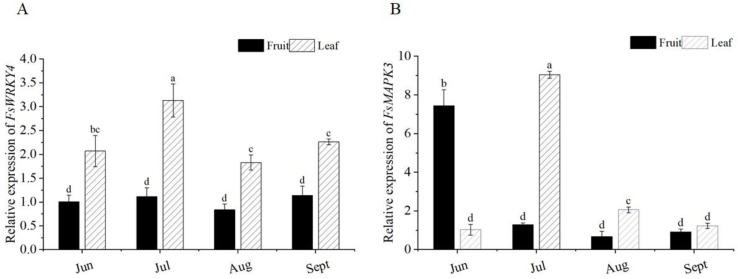
Analysis of expression levels of *FsWRKY4* and *FsMAPK3* in different stages and tissues of *F. suspensa.* The months of June to September represent four different stages. Fruit and leaf are distinct tissues. Different lowercase letters indicate significant differences between samples (*p* < 0.05). (**A**) analysis of expression levels of *FsWRKY4* in different stages and tissues of *F. suspensa*; (**B**) analysis of expression levels of *FsMAPK3* in different stages and tissues of *F. suspensa.* Data correspond to means ± SD (*n* = three biological replicates). Different letters on bars indicate significant differences according to analysis of variance followed by Turkey’s post hoc test (*p* < 0.05).

**Figure 3 plants-12-03415-f003:**
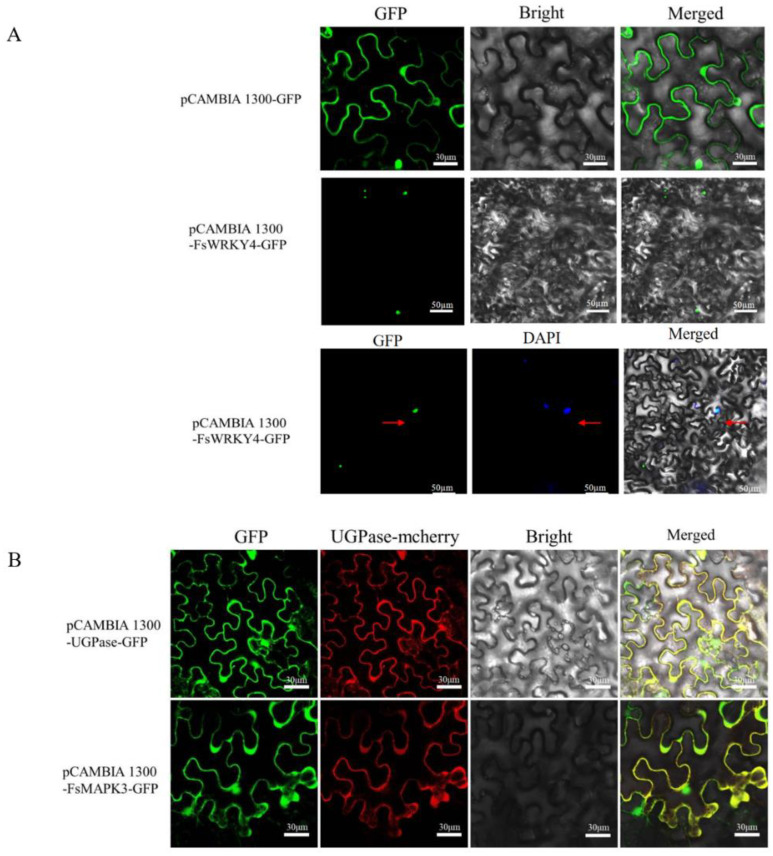
Subcellular localization of FsWRKY4 and FsMAPK3 proteins in lower epidermal cells of tobacco leaves. (**A**) Subcellular localization of FsWRKY4 protein in lower epidermal cells of tobacco leaves. “Green fluorescence” represents GFP fluorescence, “blue fluorescence” represents fluorescence emitted from nuclei stained with DAPI, “red arrow” points the nuclei with colocalization. DAPI: 4′, 6-diamidino-2-phenylindole is a blue fluorescent DNA stain that was used to indicate nuclear region. (**B**) Subcellular localization of FsMAPK3 protein in lower epidermal cells of tobacco leaves. “Green fluorescence” represents GFP fluorescence, “UGPase” is a protein specifically located in the endoplasmic reticulum, “red fluorescence” represents endoplasmic reticulum, “yellow fluorescence” is the overlap of green and red fluorescence, indicating that they were co-localized in the endoplasmic reticulum.

**Figure 4 plants-12-03415-f004:**
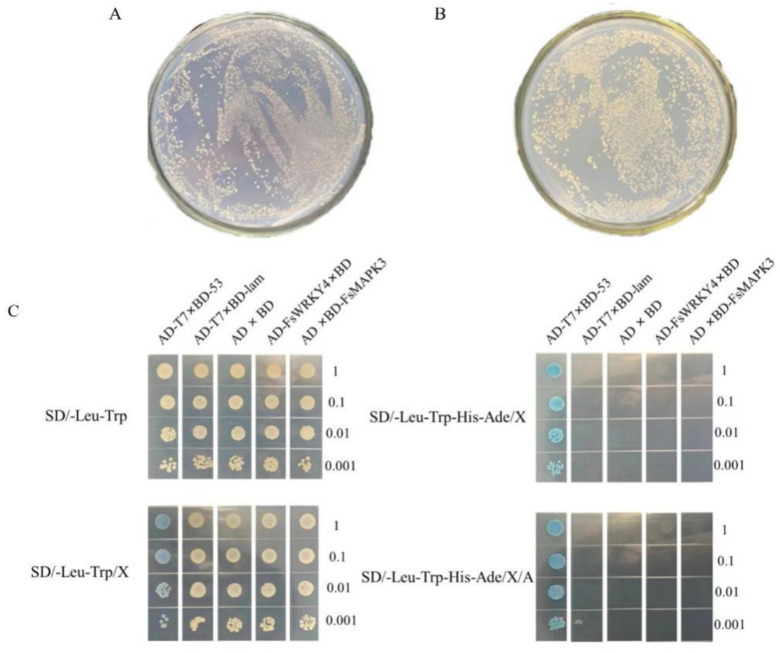
Colony and detection of colony growth by yeast two-hybrid self-activation method. Growth status of colonies transfected with the bait vectors (**A**) pGBKT7-FsMAPK3 and (**B**) pGBKT7 and placed into the yeast Y2H Gold receptor, where the strains grew normally; (**C**) detection of colony growth by yeast two-hybrid self-activation. Gradient dilution is denoted by 1, 0.1, 0.01, and 0.001. The medium contained 130 µg/mL AbA. Note: AD: GAL4 activation domain; BD: GAL4 DNA binding domain; AD-T7×BD-53: positive control; AD-T7×BD-lam: negative control; AD×BD: empty carrier control; AD-FsWRKY4×BD: recombinant prey carrier test group; AD×BD-FsMAPK3: recombinant decoy carrier test group.

**Figure 5 plants-12-03415-f005:**
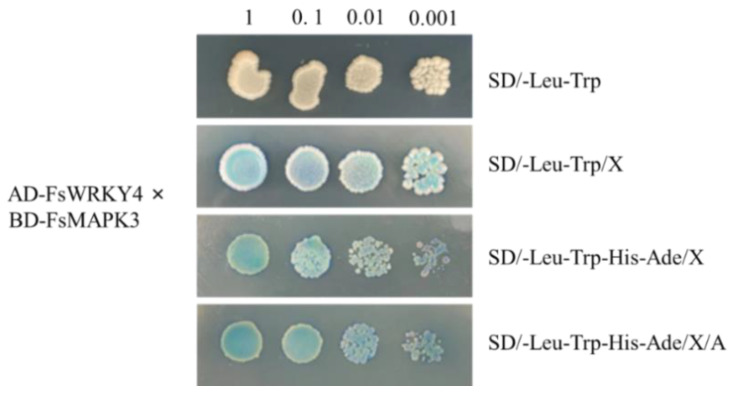
Validation of the interactions between FsWRKY4 and FsMAPK3 in a yeast two-hybrid system. The combination for this experiment was AD-FsWRKY4×BD-FsMAPK3: Y2H[pGADT7-FsWRKY4×pGBKT7-FsMAPK3]. DDO: SD/-Leu-Trp; DDO/X: SD/-Leu-Trp/X; QDO/X: SD/-Leu-Trp-His-Ade/X; QDO/X/A: SD/-Leu-Trp-His-Ade/X/A. Gradient dilution is denoted by 1, 0.1, 0.01, and 0.001. The medium contained 130 µg/mL AbA.

**Table 1 plants-12-03415-t001:** Primers for *FsWRKY4*, and *FsMAPK3* gene amplification.

Target Gene	Primer Name	Primer Sequence (5′-3′)
*FsWRKY4*	*FsWRKY4*-F	AATTCACATTCATGGCCACTAC
	*FsWRKY4*-R	CAAAACAGAAAGCTCGCTCTCT
*FsMAPK3*	*FsMAPK3*-F	TTGTCCGAATAGACAGTTGAAGC
	*FsMAPK3*-R	TAAGATGGCAGAGAAGAGTGGTG

**Table 2 plants-12-03415-t002:** Primers used for quantitative real-time PCR.

Target Gene	Primer Name	Primer Sequence (5′-3′)
*FsWRKY4*	q*FsWRKY4*-F	GCGGTGGAGAATAAGGAGGA
	q*FsWRKY4*-R	CGGAACAAACCCAGGAGAAT
*FsMAPK3*	q*FsMAPK3*-F	GACGTGTGGTCTGTAGGTTGC
	q*FsMAPK3*-R	TTCTCTTTCCGGGATTGATTG
*FsUKN1*	q*FsUKN1*-F	CAGACCAGCTTTGAGGAGTATC
	q*FsUKN1*-R	GGCCAGAAACCAGTAGTCAATA

**Table 3 plants-12-03415-t003:** Primers for subcellular localization amplification of FsWRKY4 and FsMAPK3.

Target Gene	Primer Name	Primer Sequence (5′-3′)
*FsWRKY4*	GFP-FsWRKY4-F	gcccttgctcaccatGTCGACATGGCTGAGAACCAACCTCCT
	GFP-FsWRKY4-R	cgggggactgagctcGGTACCAGAAAGTTGTGTTACCTGTTCTTCTTTG
*FsMAPK3*	GFP-FsMAPK3-F	cgggggactgagctcGGTACCATGACAGAGTATGTTGTCACCAGATG
	GFP-FsMAPK3-R	gcccttgctcaccatGTCGACGATATTTAGTGCCAAAGCCTCCTG

**Table 4 plants-12-03415-t004:** Primers for yeast two hybrid amplification of FsWRKY4 and FsMAPK3.

Target Gene	Primer Name	Primer Sequence (5′-3′)
*FsWRKY4*	AD-FsWRKY4-F	gtaccagattacgctCATATGATGGCTGAGAACCAACCTCCT
	AD-FsWRKY4-R	cagctcgagctcgatGGATCCTCAAGAAAGTTGTGTTACCTGTTCTTC
*FsMAPK3*	BD-FsMAPK3-F	tcagaggaggacctgCATATGATGACAGAGTATGTTGTCACCAGATG
	BD-FsMAPK3-R	ccgctgcaggtcgacGGATCCTTAGATATTTAGTGCCAAAGCCTCC

## Data Availability

The gene sequences of FsWRKY4 (GenBank number: OR566682) and FsMAPK3 were loaded from the National Center for Biotechnology Information (NCBI) database (FsWRKY4 GenBank number: OR566682) and (FsMAPK3 GenBank number: OR566683). Primer sequences for the FsWRKY4, FsMAPK3 gene amplification and qRT-PCR have been listed in Table 1 and Table 2, respectively.

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
