# Peer review of "Gene Expression and Interaction Analysis of FsWRKY4 and FsMAPK3 in Forsythia suspensa"

_plants, 2023, doi:10.3390/plants12193415_

Round 1
Reviewer 1 Report
MAPKs and WRKYs are reported to be associated with the accumulation of secondary metabolites. The manuscript cloned the FsWRKY4 and FsMAPK3 genes from Forsythia suspensa, examined the expression patterns of the two genes in different developmental stages in leaf and fruit, and analyzed their subcellular localizations. The interactions between FsWRKY4 and FsMAPK3 proteins was also analyzed. The English of the manuscript needs to be greatly improved. The comments for the manuscript is as follows.
1. In the Abstract section, for “1578bp and 522bp” in line 26, there should be a blank between the number and “bp”.
2. In the Introduction section, the 4th paragraph, line 63, the sentence “Since the first discovery of WRKY transcription factors in sweet potato in 1994 [14].”needs to be revised.
3. In the Introduction section, the end of 4th paragraph (line72.73), “Therefore, it is important to study the metabolic pathway of phenolpropane for the synthesis of coniferin in F. suspensa.” It is hard to draw such a conclusion. This sentence is suggested to be deleted.
4. In the Introduction section, line78-80, the sentence “When AtMAPK4 activated by light, phosphorylated modified the AtMYB75 transcription factor and promoted the accumulation of anthocyanins [24].” needs to be revised to make its meaning clearer.
5. In the Introduction section, the 7th paragraph (line 90-96) should be deleted, since yeast two-hybrid system is very well known. Thus, it is not necessary to introduce the technique.
6. In the Result section, for the caption of Figure 1, ”cDNA sequence” (line 115, 116) should be changed to ”cDNA fragment”.
7. In the Result section, for the caption of Figure 2, ”treatments” (line 130) can be changed to ”samples”, since there are no any treatments. Meanwihile, ”tissue” (line 131 and 132) should be changed to ”tissues”.
8. For the Result section 2.3, the phrase “, which is consistent with the prediction of the software” (line 143), herein the name of the “software” should be mentioned.
9. For the Result section 2.4, for the phrase “when translated and expressed in yeast” (line 156), “translated” and “expressed” are repeated in meaning.
10. For the Table 2, please check the correctness of the primer names of FsMAPK3 and FsUKN1.
11. For the Materials and Methods section 4.5, the sentence “GFP vector and ligated using homologous recombinase (Nanjing, Novozymes) to obtain pCAMBIA1300-FsWRKY4/ FsMAPK3-GFP.” (line 320, 321) should be revised to make the meaning clearer.
12. For the Reference, one of the two sets of ordinal numbers before the references should be deleted.
The English of the manuscript needs to be greatly improved.
Author Response
MAPKs and WRKYs are reported to be associated with the accumulation of secondary metabolites. The manuscript cloned the FsWRKY4 and FsMAPK3 genes from Forsythia suspensa, examined the expression patterns of the two genes in different developmental stages in leaf and fruit, and analyzed their subcellular localizations. The interactions between FsWRKY4 and FsMAPK3 proteins was also analyzed. The English of the manuscript needs to be greatly improved. The comments for the manuscript is as follows.
Response:We agree to your comment. Thank you very much! We have asked the agency(http://www.kgsupport.com/) to edit and improve the language of the revised manuscript.
Point 1. In the Abstract section, for “1578bp and 522bp” in line 26, there should be a blank between the number and “bp”.
Response 1: Thank you very much! We have left spaces between the number and “bp”. We have checked the whole manuscript and gave a blank between the number and “bp”.
Point 2. In the Introduction section, the 4th paragraph, line 63, the sentence “Since the first discovery of WRKY transcription factors in sweet potato in 1994 [14].”needs to be revised.
Response 2: We agree to your comment. Thank you very much! We have changed “ Since the first discovery of WRKY transcription factors in sweet potato in 1994” to “WRKY transcription factor was first reported in sweet potato in 1994” in the revised manuscript.
Point 3. In the Introduction section, the end of 4th paragraph (line72.73), “Therefore, it is important to study the metabolic pathway of phenolpropane for the synthesis of coniferin in F. suspensa.” It is hard to draw such a conclusion. This sentence is suggested to be deleted.
Response 3: We agree to your comment. Thank you very much! We have deleted it in the revised manuscript.
Point 4. In the Introduction section, line78-80, the sentence “When AtMAPK4 activated by light, phosphorylated modified the AtMYB75 transcription factor and promoted the accumulation of anthocyanins [24].” needs to be revised to make its meaning clearer.
Response 4: We agree to your comment. Thank you very much! We have changed “When AtMAPK4 activated by light, phosphorylated modified the AtMYB75 transcription factor and promoted the accumulation of anthocyanins [24].” to “The activation of AtMAPK4 by light, led to phosphorylated modification of the AtMYB75 transcription factor and promoted the accumulation of anthocyanins [24]” in the revised manuscript.
Point 5. In the Introduction section, the 7th paragraph (line 90-96) should be deleted, since yeast two-hybrid system is very well known. Thus, it is not necessary to introduce the technique.
Response 5: We agree to your comment. Thank you very much! We have deleted it in the revised manuscript. Thanks again.
Point 6. In the Result section, for the caption of Figure 1, ”cDNA sequence” (line 115, 116) should be changed to ”cDNA fragment”.
Response 6: We agree to your comment. Thank you very much! we have changed “cDNA sequence” to “cDNA fragment” in the revised manuscript.
Point 7. In the Result section, for the caption of Figure 2, ”treatments” (line 130) can be changed to ”samples”, since there are no any treatments. Meanwihile, ”tissue” (line 131 and 132) should be changed to ”tissues”.
Response 7: We agree to your comment. Thank you very much! we have changed “treatments” to “samples”, “tissue” to “tissues” in the revised manuscript.
Point 8. For the Result section 2.3, the phrase “, which is consistent with the prediction of the software” (line 143), herein the name of the “software” should be mentioned.
Response 8: We agree to your comment. Thank you very much! We have added the name of the “software” in the revised manuscript, which is “Cell-Ploc2.0 ( http://www.csbio.sjtu.edu.cn/bioinf/Cell-PLoc-2/)”.
Point 9. For the Result section 2.4, for the phrase “when translated and expressed in yeast” (line 156), “translated” and “expressed” are repeated in meaning.
Response 9: We agree to your comment. Thank you very much! We have deleted “translated and ” in the revised manuscript
Point 10. For the Table 2, please check the correctness of the primer names of FsMAPK3 and FsUKN1.
Response 10: We agree to your comment. Thank you very much! We am sorry. Due to our carelessness, we wrote the primer name wrong. In the Table 2, the second “qFsMAPK3-F” should be “qFsMAPK3-R” of the primer names of FsMAPK3. The the second “qFsUKN1-F” should be “qFsUKN1- R” of the primer names of FsUKN1. We have change them in the revised manuscript.
Point 11. For the Materials and Methods section 4.5, the sentence “GFP vector and ligated using homologous recombinase (Nanjing, Novozymes) to obtain pCAMBIA1300-FsWRKY4/ FsMAPK3-GFP.” (line 320, 321) should be revised to make the meaning clearer.
Response 11: We agree to your comment. Thank you very much! We have changed “GFP vector and ligated using homologous recombinase (Nanjing, Novozymes) to obtain pCAMBIA1300-FsWRKY4/ FsMAPK3-GFP.” to “The GFP vector was ligated using homologous recombinase (Nanjing, Novozymes) to obtain pCAMBIA1300-FsWRKY4/ FsMAPK3-GFP” in the revised manuscript.
Point 12. For the Reference, one of the two sets of ordinal numbers before the references should be deleted.
Response 12: We agree to your comment. Thank you very much! We have deleted
one of the two sets of ordinal numbers before the references.

Reviewer 2 Report
After reading the manuscript entitled: Gene Expression and Interaction Analysis of FsWRKY4 and 2
FsMAPK3 in Forsythia suspensa. The authors need to address the comments and concerns as below
Abstract
Comments:
1- Rebuilt the abstract with a clear introduction (clear problematic), which is suggested to show the importance of your study.
2- Reformulate the objective and methodology with clear and brief sentences.
3- In summary, mention the projection of your results for agronomic purposes.
4- Be careful with language issues,
Introduction
1. Line 40: add references
2. Add paragraph about the important and usage of Forsythia suspensa
3. Line 59: add references,
4. Please add studies that used the FsWRKY4 and FsMAPK3 genes
5. Be careful with language ( I noted many issues in the text)
Methods
Growth conditions and sampling time not clear?
Please add a separate statistics section with used tests and their details
Did you made multiple sequence alignment, phylogenetic analysis, and conserved motif construction?
Results
Please detail your results with a concentration on the most relevant findings to show the value of your study.
Discussion
The discussion is poor.
Please put your relevant findings and discuss them with references that have similar results on the other species or other aspects. Use more recent references and try to explain the link with drought.
Extensive editing of English language required
Author Response
After reading the manuscript entitled: Gene Expression and Interaction Analysis of FsWRKY4 and FsMAPK3 in Forsythia suspensa. The authors need to address the comments and concerns as below
Abstract
Comments:
Point 1- Rebuilt the abstract with a clear introduction (clear problematic), which is suggested to show the importance of your study.
Response 1: We agree to your comment. Thank you very much! We have asked the agency(http://www.kgsupport.com/) to edit the language and improve the abstract of the revised manuscript.
Point 2-Reformulate the objective and methodology with clear and brief sentences.
Response 2: We agree to your comment. Thank you very much! We have modified the objective and methodology with clear and brief sentences in the revised manuscript.
Point 3-In summary, mention the projection of your results for agronomic purposes.
Response 3: We agree to your comment. Thank you very much! In the manuscript, agronomic purposes were not involved in the manuscript, which laid a foundation for the analysis of the synthesis pathway of secondary metabolites of F. suspensa. Our next deeper research will focus on agronomic traits and ultimately achieve the goal of obtaining high-quality new varieties of F. suspensa.
Point 4- Be careful with language issues,
Response 4:We agree to your comment. Thank you very much! We have asked the agency(http://www.kgsupport.com/) to edit and improve the language of the revised manuscript.
Introduction
Point 1. Line 40: add references
Response 1:We agree to your comment. Thank you very much! We have added references in the revised manuscript.
Point 2. Add paragraph about the important and usage of Forsythia suspensa
Response 2:We agree to your comment. Thank you very much! We have added “The development of forsythia mainly focuses on medicinal, edible and cosmetic as-pects. In terms of medicine, F. suspensa has the functions of liver protection, diuretic catheterization, anti-inflammatory and analgesic [4]. In terms of consumption, forsythia can be processed into tea and has health functions, and F. suspensa seeds can also be extracted from oil and have high nutritional value [4].” in the revised manuscript.
Point 3. Line 59: add references,
Response 3:We agree to your comment. Thank you very much! We have added references in the revised manuscript. That is “Zhong, W.; Chai, Y.; Zhang, K.; Chen, X.; Lv, K.; Tao, L. Study on the Cytochrome P450s in Phenylpropanoid Metabolic Path-way. Journal of Anhui Agricultural Sciences. 2008,13,5285-5289.”
Point 4. Please add studies that used the FsWRKY4 and FsMAPK3 genes
Response 4:We agree to your comment. Thank you very much! In this paper, the FsWRKY4 and FsMAPK3 genes of F. suspensa were studied for the first time. So, we
did not introduce the FsWRKY4 and FsMAPK3 genes in the Introduction section, but we have introduced WRKY and MAPK genes of F. suspensa in our previous research(Zhang, Ji.; Liu, X.; Chen, J.; Guo, G.; Tan, X.; Zhao, X.; Xu, H.; Liu, H.; Hou, D. Screening and identification of transcription factors WRKY involved in phenylpropane synthesis pathway in Forsythia suspensa. Chinese Traditional and Herbal Drugs, 2023,18,6055-6064). We have added the reference in the Introduction section of revised manuscript.
Point 5. Be careful with language ( I noted many issues in the text)
Response 5:We agree to your comment. Thank you very much! We have asked the agency(http://www.kgsupport.com/) to edit and improve the language of the revised manuscript.
Methods
Point 1. Growth conditions and sampling time not clear?
Response 1: We agree to your comment. Thank you very much! We have specified the growth conditions and sampling time in the revised manuscript. That is “The growth of F. suspensa plants were given normal water and fertilizer conditions” and “the leaves and fruits were picked on the 10th of every monthe from May to September. ”
Point 2. Please add a separate statistics section with used tests and their details
Response 2: We agree to your comment. Thank you very much! The main content of this manuscript is the study of FsWRKY4 and FsMAPK3 genes cloning and subcellular localization, only the changes of expression of these two genes in different developmental periods involve significant analysis. We have added “4.7 Statistic analysis” in the Methods Section of revised manuscript.
Point 3. Did you made multiple sequence alignment, phylogenetic analysis, and conserved motif construction?
Response 3: We agree to your comment. Thank you very much! Based on the analysis of F. suspensa transcriptome data, we conducted WRKY multiple sequence alignment, phylogenetic analysis, and conserved motif construction in our research group has published papers(Zhang, Ji.; Liu, X.; Chen, J.; Guo, G.; Tan, X.; Zhao, X.; Xu, H.; Liu, H.; Hou, D. Screening and identification of transcription factors WRKY involved in phenylpropane synthesis pathway in Forsythia suspensa. Chinese Traditional and Herbal Drugs, 2023,18,6055-6064.). We have add the reference in the Introduction section of revised manuscript.
Results
Point 1. Please detail your results with a concentration on the most relevant findings to show the value of your study.
Response 1: We agree to your comment. Thank you very much! We have modified the results in the revised manuscript.
Discussion
Point 1.The discussion is poor.
Response 1: We agree to your comment. Thank you very much! We have modified the discussion in the revised manuscript.
Point 2. Please put your relevant findings and discuss them with references that have similar results on the other species or other aspects. Use more recent references and try to explain the link with drought.
Response 2: We agree to your comment. Thank you very much! We have modified the discussion in the revised manuscript. We also have used recent reference. Since drought stress is not mentioned in this paper, we will carry out drought stress research according to your suggestion in the follow-up research

Round 2
Reviewer 1 Report
The authors have clearly replied the comments of the reviewer, and the manuscript has been sufficiently improved. I recommend to accept the manuscript.
Reviewer 2 Report
Accept in present form